# Spittlebugs of Mediterranean Olive Groves: Host-Plant Exploitation throughout the Year

**DOI:** 10.3390/insects11020130

**Published:** 2020-02-18

**Authors:** Nicola Bodino, Vincenzo Cavalieri, Crescenza Dongiovanni, Matteo Alessandro Saladini, Anna Simonetto, Stefania Volani, Elisa Plazio, Giuseppe Altamura, Daniele Tauro, Gianni Gilioli, Domenico Bosco

**Affiliations:** 1CNR—Istituto per la Protezione Sostenibile delle Piante, Strada delle Cacce, 73, 10135 Torino, Italy; nicola.bodino@ipsp.cnr.it (N.B.); elisa.plazio@yahoo.it (E.P.); 2CNR—Istituto per la Protezione Sostenibile delle Piante, SS Bari, Via Amendola 122/D, 70126 Bari, Italy; vincenzo.cavalieri@ipsp.cnr.it (V.C.); giuseppe.altamura@ipsp.cnr.it (G.A.); 3CRSFA—Centro di Ricerca, Sperimentazione e Formazione in Agricoltura Basile Caramia, Via Cisternino, 281, 70010 Locorotondo (Bari), Italy; enzadongiovanni@crsfa.it (C.D.); danieletauro89@gmail.com (D.T.); 4Dipartimento di Scienze Agrarie, Forestali e Alimentari, Università degli Studi di Torino, Largo Paolo Braccini, 2, 10095 Grugliasco, Italy; matteo.saladini@unito.it; 5Agrifood Lab, Dipartimento di Medicina Molecolare e Traslazionale, Università degli Studi di Brescia, 25123 Brescia, Italy; anna.simonetto@unibs.it (A.S.); stefania.volani@gmail.com (S.V.); gianni.gilioli@unibs.it (G.G.)

**Keywords:** *Philaenus spumarius*, host-plant selection, plant preference, spittlebugs, xylem-sap feeders, olive, insect vectors, *Xylella fastidiosa*, OQDS, insect aggregation

## Abstract

Spittlebugs are the vectors of the bacterium *Xylella fastidiosa* Wells in Europe, the causal agent of olive dieback epidemic in Apulia, Italy. Selection and distribution of different spittlebug species on host-plants were investigated during field surveys in 2016–2018 in four olive orchards of Apulia and Liguria Regions of Italy. The nymphal population in the herbaceous cover was estimated using quadrat samplings. Adults were collected by sweeping net on three different vegetational components: herbaceous cover, olive canopy, and wild woody plants. Three species of spittlebugs were collected: *Philaenus spumarius* L., *Neophilaenus campestris* (Fallén), and *Aphrophora alni* (L.) (Hemiptera: Aphrophoridae). *Philaenus spumarius* was the predominant species both in Apulia and Liguria olive groves. Nymphal stages are highly polyphagous, selecting preferentially Asteraceae Fabaceae plant families, in particular some genera, e.g., *Picris*, *Crepis*, *Sonchus*, *Bellis*, *Cichorium,* and *Medicago*. Host-plant preference of nymphs varies according to the Region and through time and nymphal instar. In the monitored sites, adults peak on olive trees earlier in Apulia (i.e., during inflorescence emergence) than in Liguria (i.e., during flowering and beginning of fruit development). Principal alternative woody hosts are *Quercus* spp. and *Pistacia* spp. Knowledge concerning plant selection and ecological traits of spittlebugs in different Mediterranean olive production areas is needed to design effective and precise control strategies against *X. fastidiosa* vectors in olive groves, such as ground cover modifications to reduce populations of spittlebug vectors.

## 1. Introduction

True spittlebugs (Hemiptera: Aphrophoridae), a family of xylem-sap feeder insects, recently gained fame as the vectors of the exotic plant pathogenic bacterium *Xylella fastidiosa* Wells in Europe [1,2,3]. The introduction in Europe of this xylem-limited bacterium, the causal agent of the Olive Quick Decline Syndrome (OQDS), led to dramatic dieback of olives in Apulia in recent years and prompted applied research on spittlebugs, previously overlooked as agricultural pests in Europe [4]. Currently, several aspects of bionomics of Aphrophoridae, e.g., abundance, spatial and ecological distribution, temperature developmental threshold, and alternative host plants, are not yet well known, particularly in cropping systems, such as the olive groves in the Mediterranean area. Recent field surveys in olive groves of different Mediterranean countries have shown the predominance of Aphrophoridae as xylem-sap feeder group in the Mediterranean environment, with little or no presence of sharpshooters (Hemiptera: Cicadellidae: Cicadellinae) [5,6,7,8]. These findings highlighted the profound difference between *X. fastidiosa* pathosystems in Europe compared to the ones occurring in the United States of America and Brazil, where sharpshooters are the predominant xylem-sap feeders and main vectors of the bacterium [9,10,11].

Spittlebugs are among the main phytophagous insects for biomass in a meadow environment in temperate regions of Europe, although presenting a relatively low number of species [12,13,14,15]. European spittlebugs are generally univoltine, i.e., one generation per year, and their five nymphal instars are characterized by the production of foamy masses on herbaceous or woody plants, with shelter function against desiccation, solar radiation, and predation [16,17,18]. Spittlebug adults are characterized by extraordinary jumping ability [19] and feed on xylem vessels of both herbaceous and woody plants [2,20,21].

*Philaenus spumarius* L. (Hemiptera: Aphrophoridae) is the most ubiquitous and abundant spittlebug species in Europe, and is, therefore, considered the key vector in Apulia epidemic of *X. fastidiosa* [1,3,22,23]. In Mediterranean conditions, nymphs develop through spring on herbaceous plants, and adults start to emerge in late April–May [4,8,24,25]. Upon emergence, adults move onto woody plants in late spring/early summer also as a result of the water stress of herbaceous cover [4,24]. Adults tend to return on herbaceous cover in late summer or early autumn, where females oviposit eggs that will hatch the following spring [6,16,25]. A schematic representation of the spittlebug life cycle in olive agroecosystems can be found in European Food Security Agency (EFSA) [26].

Most spittlebugs are oligophagous or polyphagous [12,27,28], and feed on a wide range of host-plants apparently because xylem sap is relatively free from chemical defenses common in phloem sap and other plant parts [29,30]. The meadow spittlebug *P. spumarius* is considered one of the most polyphagous insects, with at least 500 documented host-plants [4,12,16,31]. However, data on *P. spumarius* host-plant association are mainly qualitative information on plant selection among its host range [12,32,33] and only preliminary information is available in agroecosystems, including those affected by *X. fastidiosa* in Europe [6,8]. The genus *Philaenus* is considered to be originally monophagous during nymphal stages on asphodel plants (*Asphodelus* L.), and the adaptation to various plant food sources made possible a great expansion of *P. spumarius* distribution [24,34]. Indeed, the monophagous species *P. italosignus* Drosopoulos & Remane—a reported vector of *X. fastidiosa*—is restricted to areas and groves where asphodel is abundant [35,36]. Conversely, the related species *Neophilaenus campestris* (Fallén) can reach quite high population levels in olive groves, and in Spain has been often reported as the most abundant species [6,37]. Its nymphal stages are usually associated with Poaceae, whereas adults tend to estivate on coniferous trees [37,38].

A detailed description of host plants and their preferential exploitation by both nymphal and adult stages of spittlebugs in the Mediterranean region is still lacking. Indeed, from the perspective of *X. fastidiosa* control, it is essential not only to identify the wild host-plants that allow nymphal development but also to determine the most important host-plants that permit the build-up and establishment of abundant adult populations of spittlebugs, thus speeding up the spread of the bacterium.

The aim of our study is to quantitatively describe the selection and distribution of nymphal and adult stages of spittlebugs on both wild and crop plants in olive groves during their entire life cycle. Data were collected during a three-year survey of spittlebug populations in olive agroecosystems in two regions of Italy—Apulia and Liguria—presenting different climate characteristics within the Mediterranean climate. Our results provide different information on spittlebugs bionomics: (1) identification of key plant taxa that, both within and nearby olive groves, allow nymphal development and adult survival of spittlebugs and can act as reservoirs of the vectors’ populations; (2) description of exploitation of host-plants by nymphs based also on their instar and plant phenology; (3) host-plant selection of *P. spumarius* nymphs related to the abundance of plant taxa in the sampling sites; (4) patterns of spatial distribution of nymphal stages. This information will help (i) to draw up effective and selective management strategies against spittlebugs in Mediterranean regions, (ii) to estimate risks of *X. fastidiosa* outbreaks in non-infectedareas based on plant composition of agroecosystems.

## 2. Materials and Methods

### 2.1. Survey Sites

Four olive groves were surveyed in two Italian Regions, Apulia (about 41° N) and Liguria (about 44° N), during three years of field sampling (2016–2018) to assess host-plant exploitation of xylem-sap feeder insects. The monitored olive groves in Apulia were outside the *X. fastidiosa*-infected area at the time of the survey. Olive groves were selected based on low-input management, e.g., no insecticides were sprayed, no tillage was applied in the olive groves during the study period in Northern Italy, while soil was plowed in Southern Italy in June or July—thus not affecting the development of spittlebug nymphs—to avoid fire hazards during summer. In each Region, the selected olive groves were located one in a coastal area and the other one in an inland area, to investigate spittlebug populations under different climatic conditions within the Mediterranean basin. Detailed information regarding the characteristics of surveyed olive groves is reported in Bodino et al. (2019) [25]. From 2017, the field survey in the inland olive grove in Apulia (Locorotondo municipality) was moved to a new olive grove, very close to the previous one, because the grower tilled the soil and sowed broad bean planting in the olive grove monitored during 2016. Therefore, surveys in the two Apulia olive groves only were prolonged to 2018. The sampled area in each survey site was a homogeneous area of 1 ha (i.e., primary sampling unit: PSU), including both the agricultural crop (olive) and alternative woody host-plants (e.g., hedgerows of native garrigue plant species).

### 2.2. Sampling of Nymphal Stages

Nymphal populations of spittlebugs and their host-plants were monitored within the PSU using the quadrat sampling method, frequently used to quantify the abundance of spittlebug nymphs [6,8,39,40]. Thirty quadrats (secondary sampling unit for preimaginal instars: SSU_p_; 0.25 m^2^—100 cm × 25 cm each) were randomly positioned on the ground cover inside each PSU. Since sampling is partially destructive, the SSU_p_ was always different within and between sampling dates. Vegetation and soil surface inside the quadrat were visually inspected for the presence of spittlebug nymphs. These latter were identified at the species level and their preimaginal stage determined. The number of nymphs per spittle was assessed per individual host-plant and per host-plant taxon, together with the position of the nymphs on the individual host-plant (bottom, medium, and upper third). Spittlebug samplings were conservative, and instar was determined directly in the field using classification from Vilbaste (1982) [41]. Only a few nymphs from each olive groves were collected and reared to the adult stage in laboratory facilities to confirm species identification.

The ground plant community was defined for each sample quadrat based on (i) average height of herbaceous plant cover, (ii) total plant cover percentage, and (iii) cover percentage of the most abundant plant taxa (range: 2–12 dominant plant taxa/SSU_p_). Plants found with spittlebug foams (i.e., host-plants) and most abundant plants inside SSU_p_ were identified at the genus level directly in the field or sampled to be later identified in the laboratory using classification keys from Pignatti (1982) [42]. Since field samplings started in late winter, often plants were not showing yet characteristic morphological traits necessary for correct identification at the species’ level. The phenological phases of host-plants at the time of sampling have been classified into three classes: the pre-flowering, flowering, and post-flowering phase. Sampling of nymphal stages was carried out weekly or fortnightly from early March until Late-May or until no nymphs were found in any SSU_p_ for two consecutive sampling dates.

### 2.3. Sampling of Adult Stage

Spittlebug adults were sampled using a sweep net (38 cm in diameter) on three different vegetation types inside each olive grove: herbaceous cover, olive trees, and wild woody plants (shrubs or trees growing inside or in the immediate surroundings of the olive orchard). Sweep net has proven to be the most reliable sampling method for the adult stage of spittlebugs [6,43]. Samplings were carried out at random locations within PSU for each sampling date to avoid repeated disturbance on the same points during the sampling season. Spittlebugs in the ground vegetation were sampled in 30 randomly distributed secondary sampling units (SSU_h_), consisting of 4 sweeps each performed along a 2.8-m transect (120 sweeps per site per sampling dates). Therefore, the area effectively sampled by sweeping in a SSU_h_ was estimated in about 1.0 m^2^. (0.7 m length × 0.38 m width × 4 sweeps). On olive canopies, insects were sampled using a sweep net (38 cm in diameter) with a 2-m long stick on 20 randomly distributed olive trees (SSU_o_); 10 sweeps were performed on each SSU_o_, spread around the entire olive canopy. Samplings on wild woody plants were carried out with the same methodology used for olive trees, but on 10 different plants randomly chosen (SSU_s_). Given the dense Mediterranean shrubland (garrigue), it was not always possible to perform the sweeps all around a single plant. Wild woody plant species present in each site were identified to species level using Pignatti (1982) [42]. Phenology of sampled olive trees was assessed through visual observation, following the BBCH scale [44]. A sampling of spittlebug adults was carried out from their first appearance in quadrat samplings until late autumn or when no spittlebugs were found in any SSU for two consecutive dates. Spittlebug adult samplings were conservative, thus collected spittlebugs were immediately released in the field after being identified.

### 2.4. Statistical Analysis

To investigate if the number of spittlebug nymphs (response variable) was different according to cover percentage (continuous) and mean height (continuous) of herbaceous cover (covariates), a negative binomial generalized linear mixed model (GLMM) was performed. Distribution of nymphs on host-plants (response variable: no. of nymphs) according to the covariates’ Spittle position (part of host-plant on which the spittle was found), host-plant phenology, and Nymphal instar was analyzed as a Poisson GLMM in which year, region, olive grove, and sampling date were used as random intercepts. The models assumptions were verified by plotting residuals versus fitted values versus each covariate in the model. The package lme4 in the software R [45] was used to fit the models.

To assess the host-preference of *P. spumarius*, we compared the relative frequency distribution of plant taxa in the sampling area (V) with the relative frequency distribution of plant taxa selected by nymphs of *P. spumarius* (S). V(i) is the proportion of SSU_p_ covered by the *i*-th plant taxon (i=1,…k). S(i) is computed as the proportion of individuals found on the *i*-th plant taxon out of the total number of *P. spumarius* individuals found in the sampling area. The hypothesis is that in the absence of host-preference by *P. spumarius*, S should be equal to V. For the *i*-th plant taxon, if S(i)>V(i), this plant taxon is more attractive for spittlebugs than the others, otherwise if S(i)<V(i), the plant taxon is less attractive for *P. spumarius* than the others. To measure the degree of deviation between S and V we applied the Kullback–Leibler divergence ($D_{KL}$) measure [46]:$$D_{KL}\left(S\|V\right)=\sum_{i=1}^{k}S\left(i\right)\ln\left(\frac{S\left(i\right)}{V\left(i\right)}\right);$$
$$KLD\left(i\right)=S\left(i\right)\ln\left(\frac{S\left(i\right)}{V\left(i\right)}\right);\;\;KLD\left(i\right)\in \left(-\infty ,\;+\infty \right).$$

KLD(i) represents the level of attractiveness of the *i*-th plant taxon for *P. spumarius.*

Host-preference analyses were conducted for each site separately, as the plant compositions found in the 4 survey sites are quite different, given their geographical and altitudinal diversity. At each sampling site, a two-step analysis was performed. In the first step, we analyzed the host-plant preferences at plant family taxon. We defined $\bar{V}$ as the mean over time and sampling unit of the relative frequency distribution of plant species in the sampling site and $\bar{S}$ as the mean over time and sampling unit of the relative frequency distribution of plant species selected by *P. spumarius* ($\bar{S}$). $\bar{V}\left(i\right),$ and $\bar{S}\left(i\right)$ have been used to compute the Kullback–Leibler divergence. In the second step, the analysis was performed on a genus level for each plant family separately. We focused on the two plant families on which more *P. spumarius* individuals were sampled. In this step, V(i) represents the proportion of area covered by the *i*-th plant genus over the area covered only by the considered plant family. S(i) is computed as the proportion of individuals found on the *i*-th plant genus out of the total number of *P. spumarius* individuals found on the considered plant family. To compute KLD(i), we calculated the mean over time and sampling unit of V(i) and S(i). The analysis of host-preference was conducted on nymphal stages’ sampling data only. Indeed, adult stage association with specific plants is more difficult to assess, since adults can be sampled on plants on which they do not feed upon, given their high mobility; moreover, sampling through sweep net did not allow specific plants to be sampled, especially on herbaceous cover.

Aggregation of nymphs of spittlebugs on single plants and spittles was described by the mean of nymphs/plant and nymphs/spittle, together with 95% confidence intervals of sample calculated for each spittlebug species through bootstrapping procedure (10,000 iterations, adjusted bootstrap percentile (BCa) method), using the R package *boot* [47]. Aggregation of nymphal instar of *P. spumarius* in sampling quadrats and within spittles was assessed using Iwao mean-crowding function. Index of mean crowding ($\dot{x}$), i.e., “the mean number per individual of other individuals in the same quadrat” [48], is given by:$$\dot{x}=\bar{x}+\left(\frac{s^{2}}{\bar{x}}-1\right),$$
where $\bar{x}$ is the sample mean of the density and $s^{2}$ is the sample variance of the density. Thus, the index $\dot{x}$ is the mean number of other individuals per sample unit per individual and is an expression of the intensity of interaction between individuals [49]. Mean crowding can be expressed over a range of densities by a linear regression [50]:$$\dot{x}=\alpha +\beta \bar{x}.$$

The intercept *α* represents the tendency to crowding (or repulsion), also referred to as “index of basic contagion” [49]. The regression slope *β* is related to the way insects spread in their habitat, named also “density contagiousness coefficient”, expressing the extent to which individuals are contagious at higher densities. The slope is equal to 1 in a random Poisson distribution, is greater than 1 in a contagious or clumped distribution and is lower than 1 in a more uniform distribution than a Poisson distribution. The same interpretation of “patchiness” index values—i.e., the ratio between mean crowding and mean density ($\frac{\dot{x}}{\bar{x}}$)—is assumed [51].

## 3. Results

### 3.1. Nymphal Stages

Nymphal stages of spittlebugs were observed on the herbaceous cover of olive groves from early March through late May. The development of nymphs onto herbaceous cover usually started and peaked slightly earlier (one-two weeks) in Apulia than in Liguria. Three spittlebug species were collected during visual observation of herbaceous vegetation: *P. spumarius*, *N. campestris* and *Aphrophora alni* (Fallen), the latter sampled in Liguria olive groves only. *Philaenus spumarius* was the most abundant species in all the surveyed sites. A detailed description of population structure through season and phenology of preimaginal instars is reported in Bodino et al. (2019) [25]. Density of *P. spumarius* nymphs increased slightly, although significantly, at higher percentages of herbaceous cover [Negative binomial GLMM: *β* = 0.004 (CI 0.001–0.007), F = 6.56, df = 1, *p* = 0.012]. No effect of height of herbaceous cover on density of *P. spumarius* nymphs was registered [Negative binomial GLMM: *β* = 0.002 (CI −0.003–0.008), F = 0.89, df = 1, *p* = 0.373].

#### 3.1.1. Selection and Exploitation of Host-Plants

The nymphs of the three species of spittlebugs, although all of them somehow polyphagous, showed several differences in plant association. Considering the plant family level, *P. spumarius* was found mainly on Asteraceae (4538 nymphs: representing 48.2% of total individuals collected) and Fabaceae (2794 nymphs: 29.7%), with other botanical families far less represented, e.g., Apiaceae (3.6%), Poaceae (3.2%), and Plantaginaceae (2.4%) (Figure 1 and Appendix A). The host-plant association with *P. spumarius* differed between olive groves located in Apulia and Liguria. Asteraceae and Fabaceae were both highly attractive to *P. spumarius* nymphs in Apulia (KLD = 0.29–0.46), whereas Asteraceae were much more attractive (KLD = 0.8–0.9) than Fabaceae (KLD = 0.02–0.03) in Liguria olive groves. Poaceae exhibited a low level of attractiveness in both Regions (KLD = −0.07–−0.18). The number of nymphs on both Apiaceae and Plantaginaceae is proportional to the percentage of cover of these plant families (KLD ≈ 0) (Figure 2). The nymphs of *N. campestris* were instead found almost exclusively on Poaceae (1771 nymphs: 95.6%), and marginally on other plants, e.g., Apiaceae (1.6%) and Asteraceae (1.4%) (Figure 1). Plant family association of *A. alni* nymphs was similar to the one of nymphs of *P. spumarius*—40.1% were found on Asteraceae, 19.9% on Fabaceae, and 12.5% on Apiaceae—although the abundance was lower, i.e., 297 nymphs found in total.

Focusing on plant genera, nymphs of *P. spumarius* were found on a total of 98 different plant genera, 42 in Apulia and 84 in Liguria (Appendix A). Most of the host-plant genera belonged to Asteraceae—19 genera in Liguria and 12 in Apulia—followed by Fabaceae—12 genera in Apulia and 8 in Liguria. Selection of host-plant genera by *P. spumarius* nymphs was greatly variable among sampling years, locations, and Regions, as shown by both insects counts and Kullback–Leibler divergence results (Figure 3 and Figure 4). *Medicago* was the plant genus with more *P. spumarius* nymphs collected on (15% on the overall data), and this host-plant genus alone accounted for 34.3% of total nymphs sampled in Apulia. However, being also the most common plant genus within the samplings (up to 36%), the KLD index was negative, showing a similar attractiveness of *Medicago* compared to other common plant genera of Fabaceae present in the studied olive groves (Figure 4). In Liguria, several Asteraceae genera (*Picris, Crepis, Sonchus, Bellis, Cichorium*) and *Medicago* were highly attractive in both olive groves (Figure 3 and Figure 4). Interestingly, the Asteraceae genus *Hyoseris* hosted 4.4% of total *P. spumarius* nymphs but, being very common in Liguria olive groves, the KLD index was strongly negative, showing a low attractiveness of this genus. Plant genera *Cervaria* (Apiaceae), *Plantago* (Plantaginaceae), *Euphorbia* (Euphorbiaceae), and *Stellaria* (Caryophyllaceae) appeared highly attractive only in the second year of the study in Liguria (Appendix A).

Nymphal stages of *N. campestris* were found mainly on Poaceae (Figure 1), with genera *Poa*, *Arrhenatherum*, *Anisantha,* and *Hordeum* more represented. It is worth noting that grasses were only seldom identified at genus level, given the absence of morphological traits necessary for identification in early spring. Thus, we do not provide precise nymph/host-plant association data at the genus level. Only occasionally *N. campestris* nymphs were found on forbs, e.g., genera *Crepis*, *Hyoseris,* and *Medicago* (4.2% of total nymphs) (Figure 1). *Aphrophora alni* nymphs were observed on 37 different plant genera, although the genera *Daucus*, *Trifolium*, *Picris*, and *Sonchus* hosted 37% of total *A. alni* nymphs (Figure 1).

#### 3.1.2. Selection of Host-Plant Taxa during Time

Host-plant selection by *P. spumarius* nymphs changed over time. Species belonging to Asteraceae (mainly genera *Bellis* and *Picris*) were mainly exploited at the beginning and the end of the nymphal development in Liguria (Figure 5). In Apulia species belonging to Asteraceae showed some peaks of attractiveness, with different temporal patterns across year and sites. Fabaceae, the dominant host-plant family in Apulia, show a high level of attractiveness in the middle-end of the season (Figure 5). The proportion of nymphs slightly increased on Poaceae in late spring in Apulia. Interestingly, considering the morphology of the host-plants, 54.9% of *P. spumarius* nymphs (i.e., 5160 individuals) were found on plants with basal rosette, although this morphological trait was present in only 23.1% of host-plant taxa identified. Both *N. campestris* and *A. alni* nymphs did not show clear differences in host-plant taxa selection during the season.

#### 3.1.3. Distribution of Nymphal Instars on Host-Plants

The recorded changes in host-plant selection by *P. spumarius* nymphs reflect differences in plant preference among preimaginal instars, as well as modifications in plant communities during the season. Early instars (i.e., 1st and 2nd) were more abundant on Asteraceae than older stages, both in Apulia (33.6% of total nymphs) and in Liguria (49.4%), compared to the percentage of early instar found on other plant families (mean = 20.3%) (Figure 1). Different Asteraceae genera hosted high percentages of early instars, such as *Bellis* (76.7% of the nymphs belongs to 1st and 2nd instars), *Cichorium* (64.9%) and *Picris* (56.4%) in Liguria and *Picris* (54.5%) and *Crepis* (58.3%) in Apulia (Figure 6). Conversely, Poaceae, Polygonaceae, Plantaginaceae, and Fabaceae hosted higher proportions of late nymphal instars of *P. spumarius* (i.e., 4th and 5th) than early nymphal instars, up to 63.1% of 5th instar on Polygonaceae in Liguria and 43.8% on Poaceae in Apulia. Late instars represented 69.6% and 54% of total nymphs on the Fabaceae genera *Medicago* and *Trifolium* in Apulia, and 60.2% and 43.5% on *Plantago* and *Crepis* in Liguria, respectively. *Neophilaenus campestris* nymphs were found almost exclusively on Poaceae regardless of the instar (Figure 1). In addition, *A. alni* did not show relevant plant association changes among nymphal stages.

#### 3.1.4. Nymphal Stages Aggregation

The mean number of spittlebug nymphs per individual host-plant was different among spittlebug species. Overall, more *P. spumarius* nymphs per plant were observed [mean = 1.79 nymphs/plant (95% CI 1.75–1.85)] than *N. campestris* [1.33 (CI 1.29–1.38)] and *A. alni* nymphs [1.49 (95% CI 1.35–1.69)]. Some herbaceous plant genera hosted significantly more *P. spumarius* nymphs per individual plant than the overall mean number of nymphs per plant, e.g., *Cervaria* (4.92 ± 0.87 nymphs/plant), *Cytisus* (4.09 ± 0.95), *Picris* (3.11 ± 0.393), *Urospermum* (2.83 ± 0.44), and *Crepis* (2.37 ± 0.15) in Liguria, *Picris* (3.11 ± 0.39) and *Vicia* (3.05 ± 0.77) in Apulia. In some cases, more than 20 *P. Spumarius* nymphs were present on a single plant at a given moment, i.e., on genera *Picris*, *Stellaria*, *Cervaria*, *Plantago*, *Onobrychis*, and *Crepis*.

The mean number of nymphs per spittle was greater for *P. spumarius* [mean = 1.45 nymphs/spittle (95% CI 1.42–1.48)] and *A. alni* [1.39 (95% CI 1.27–1.54)] than for *N. campestris* [1.24 (95% CI 1.21–1.28)]. Plant genera that hosted more *P. spumarius* nymphs often had more crowded spittles, e.g., in Liguria *Cervaria* (2.4 ± 0.28 nymphs/spittle), *Bellis* (1.95 ± 0.10), and *Taraxacum* (1.96 ± 0.26), in Apulia *Picris* (2.74 ± 0.34). The maximum number of *P. spumarius* nymphs-per-spittle was 39 (on *Picris* in Apulia), while a maximum of 8 and 7 nymphs-per-spittle was recorded for both *N. campestris* and *A. alni* on Poaceae.

The majority of *P. spumarius* spittle masses contained a single nymph (83.8%). Mean crowding—i.e., the mean number per individual of other individuals in the same quadrat—was similar among regions (Apulia = 0.774; Liguria = 0.780), but with a considerable difference among years in Apulia (Table 1). Crowding was higher in Apulia during 2017, mainly due to a few quadrats presenting a very high mean density of nymphs-per-mass (i.e., strong outliers). Distribution analyses of nymphs per spittle by Iwao’s regression with data pooled for each sampling quadrat are reported in Figure 7 and Table 1. Following preliminary analysis, four strong outlier data with mean density nymphs-per-mass > 5 were discarded. *p*-values for Iwao’s regression were not significant for samplings in Apulia during 2016, together with the low value of adjusted R^2^, therefore, results from Apulia 2016 samplings are not considered hereafter. Intercept values for Iwao’s models were negative, but not significantly different from zero, suggesting that nymphs emerging from egg masses tend to disperse randomly as individuals rather than in groups, initially creating single foam masses [49,50]. Iwao’s regression slopes were all significantly positive, showing a tendency of *P. spumarius* nymphs to aggregate in more crowded spittles as the average density per spittle increases (i.e., overdispersal). However, the aggregation pattern could also be driven by time and/or nymphal stage. For example, sampling dates showing the highest densities (and hence, the highest crowding) were in April, when the nymphal population peak was registered and mainly 3rd–4th and 5th instar were found [25]. Furthermore, early nymphal instars showed higher aggregation, e.g., in Liguria 60.8% of 1st instar and 56.5% of 2nd instar lived in spittles with ≥ 2 nymphs, whereas the percentage of nymphs living aggregated decreased to 50%–52% for later instars. In Apulia, the percentage of nymphs living aggregated was generally lower—even if the behavior is quite different between instars—45.7% of 2nd instars and 44.3% of 3rd instar lived in spittles with ≥ 2 nymphs, compared to 24.6% of 4th and 35.2% of 5th instar.

Crowding of *P. spumarius* nymphs-per-quadrat (i.e., mean number of other individuals per quadrat per individual) was twice in Liguria (9.23) than in Apulia olive groves (4.03), whereas patchiness was similar (Liguria: 1.32; Apulia: 1.17), although with great differences among sampling years (Table 1). Iwao’s regression slopes were all significantly positive, showing a tendency of *P. spumarius* nymphs to aggregate in more crowded areas as the average density per quadrat increases (i.e., overdispersal) (Appendix A).

#### 3.1.5. Spittle Position vs. Phenology of the Plant and Preimaginal Instar

Nymphs of Aphrophoridae were not randomly distributed along the height of host-plants. The abundance distribution of *P. spumarius* nymphs on host-plants was affected by position on plant height, plant phenology, and nymphal instar, with significant interaction among these three variables (Table 2). Indeed, early instars (i.e., 1st–2nd) of *P. spumarius* were observed mainly in the basal part of the host-plants (64.4% of total early instars) and on pre-flowering plants, whereas from 3rd instar nymphs were found in similar numbers along the entire height of host-plants, without clear selection for pre-flowering plants (Figure 8). However, it is not possible to separate the effect of a change in preference of nymphs and the increase in the number of flowering plants in late spring. Both *N. campestris* and *A. alni* were instead found mainly at the basal third of host-plants (respectively, 88% and 99% of total), with no effect of the nymphal stage (Figure 8).

### 3.2. Adults

Spittlebug adults were present from late April in Apulia and early May in Liguria olive groves. All the three spittlebug species—*P. spumarius*, *N. campestris* and *A. alni*—found during samplings for nymphs were collected as adults inside the olive groves, and a single individual of the Aphrophoridae *Lepyronia coleptrata* (L.) was collected on ground cover in Liguria. Most of the adults were collected on the herbaceous cover during late spring (>50% in both regions), while in summer months (July–August), adults were collected mostly on olive and alternative woody host-plants (Figure 9). In Apulia, few spittlebugs were collected in the whole olive agroecosystem during summer months (in total ≈ 30–40 adults sampled in July–August for each sampling year), whereas in Liguria the number of spittlebugs collected constantly remained high also during summer (Figure 9). In September–October spittlebugs returned to the herbaceous layer, although in Apulia, the population abundance was much lower than Liguria, probably because of different weeds’ management carried out in the two regions (e.g., mowing or tilling to reduce fire hazard in Apulia). Spittlebug adults disappeared in the olive agroecosystem from November–December, although in Apulia, few individuals may overwinter and die in early spring [25].

#### 3.2.1. Distribution on Woody Plants—Olive vs. Wild Shrubs/Trees

The number of spittlebug adults on wild woody plants located within or on the hedgerows of monitored olive varied greatly along the season, presenting higher densities during May–June in Apulia and late June through September in Liguria (range 1–5 spittlebugs/SSU_h_) (Figure 10 and Figure 11). Most spittlebug adults—i.e., 71.8% and 67% in Liguria and Apulia, respectively—were collected on *Quercus ilex* L., *Pistacia lentiscus* L., and *P. terebinthus* L., among the most common wild woody plants in Mediterranean olive agroecosystems in Italy. Nonetheless, other woody plants may host high number of spittlebugs: In Liguria, up to 5 *P. spumarius* were collected on a single SSU_s_ on *Quercus* × *crenata* Lam. and up to 4 *P. spumarius* on a single SSU_s_ on *Hedera helix* L. *Neophilaenus campestris* was only occasionally collected on woody plants, probably because only broadleaf woody plants were located in the study sites, and *Neophilaenus* spp. are known to move onto coniferous plants during drought periods [24,38]. *Aphrophora alni* reached densities similar to those of *P. spumarius* on *Quercus* spp. and *Pistacia* spp.; on a single plant of *Rhamnus alaternus* L. in 2016, up to 8 individuals on a single date was recorded (Figure 11).

#### 3.2.2. Abundance of Spittlebugs on Olive According to Crop Phenology

The highest number of spittlebugs on olive trees in Apulia was observed in May, immediately after the emergence, corresponding to the olive phenological stage of inflorescence emergence and development. During the following months, the population abundance continued to steadily reduce, reaching zero in October or November, across the late maturation of fruits and senescence phenological stages of olive (BBCH 85–92) (Figure 12). The situation was different in Liguria olive groves: The peak of spittlebugs on olive trees occurred later than in Apulia (June–July)*,* corresponding to flowering and beginning of fruit development, and the reduction was less abrupt than the one registered in Apulia. Indeed, *P. spumarius* adults were present in quite high numbers during the entire spring–summer period, especially during 2016 (Figure 12). *Philaenus spumarius* was the dominant spittlebug species on olive trees, except in Apulia during 2018, when its populations were low and similar to those of *N. campestris*. In Liguria, *A. alni* was an important component of the spittlebug community on olive, accounting for up to 20%–30% of total spittlebugs collected until July, especially in 2016.

## 4. Discussion

*Philaenus spumarius* is an extremely polyphagous insect, with more than 500 plant species recorded as hosts [16,31]. However, few studies investigated in detail the host-plant range of *P. spumarius* and other spittlebugs, especially in Europe, where they seldom caused significant crop losses [12,52,53]. The situation changed following the *X. fastidiosa* outbreak in South Italy (Apulia), and the subsequent identification of spittlebugs as the vectors of the bacterium in Europe [1,22]. Soil tilling or mowing are currently mandatory control measures of spittlebugs’ populations in infected and buffer zones in Apulia, carried out to limit the expansion of *X. fastidiosa* in South Italy [54,55]. However, the capability of spittlebugs to feed and live on many different plant species makes it difficult to effectively suppress the population of these insects. Nymphal stages may grow outside the olive groves and subsequently reach olive trees as adults; spittlebug adults can also actively disperse among host-plants surrounding the agroecosystems, spreading *X. fastidiosa* within a hidden compartment (reservoir), making even more difficult to detect and control the pathogen [56,57,58].

We conducted three-year monitoring of host-plant exploitation by spittlebug nymphs and adults in Italian olive groves. Our study better defines variation in host-plant exploitation by spittlebugs among insect species, plant taxa, phenological stage, period of the year, and insect instar. This information may guide the development of effective control strategies against spittlebugs, e.g., through modification of plant communities within and surrounding olive agroecosystems to limit their attractiveness to spittlebugs.

*P. spumarius* nymphs and adults were confirmed to be highly polyphagous, with a total of 98 plant genera exploited by nymphal stage, and adult collected on 16 woody host-plant species. Some plant genera, however, demonstrated to be of key importance for hosting high numbers of *P. spumarius* in olive agroecosystems. Asteraceae and Fabaceae were the botanical families more selected by nymphs, as also noticed in our previous study carried out in South Apulia (Lecce Province) [8]. The exploitation of plant taxa was not constant throughout the year, both for nymphs and adults. Nymphs of *P. spumarius* shifted from Asteraceae with basal rosettes (e.g., genera *Picris* and *Bellis*) to Fabaceae or Plantaginaceae (e.g., genera *Medicago* and *Plantago*), especially in Liguria. These findings may be caused partly by different phenological stages through the season and different availability of plant taxa during the season. However, preferential selection of plants based on their morphology by different instars of *P. spumarius* is another explanation. Indeed, the host-plant is a place in which the insect herbivore lives, not only a source of food [49]. Nymphs of *P. spumarius* prefer to feed in plant axils over other sites, probably because they represent higher-quality shelter, characterized by lower dispersion of the spittle, and thus an increase in the average foam size [59]. Furthermore, the apparent preference for host-plants with basal rosettes could be favored by the reduced loss of foam by evaporation in the lower parts of vegetation [17]. As the season progresses, the hardening of plant tissue below the terminal bud as the herbaceous host grows may cause difficulties for stylet penetration, compelling spittlebug nymphs to move up on the stem to feed [60]. Preferential host-plant selection could also be affected by different amino acids in the xylem sap, as partially demonstrated by Horsfield (1978) [20], although he analyzed sap from leaves, whereas *P. spumarius* feed mainly on stem or petioles.

Preferential selection of some host-plant taxa by *P. spumarius* nymphs is also supported by the aggregation pattern. Nymphs tended to concentrate in more crowded spittles and quadrats as overall density increased. This can be explained by (i) the presence of preferred plant species, (ii) attractiveness of more favorable habitat niches (e.g., basal rosettes), (iii) benefits of gregarious feeding [61]. However, since spittlebug nymphs have limited movement capability, patterns in oviposition preference by females of the previous year generation could partly cause the spatial distribution of juveniles in the agroecosystem [62,63]. Aggregation of nymphs occurring mainly on some plant species, or in specific areas within the olive agroecosystem, could favor the application of targeted control measures, e.g., mowing, provided that they are carried out with a correct timing [8,54].

Spittlebug adults were more abundant on olive trees and in the whole olive agroecosystem soon after emergence in May/June. Indeed, the peak of spittlebug abundance on olive trees occurred during the phenological stages of inflorescence emerging and development of flowers, when high content of amino acids is present in xylem, while amino acids are strongly reduced in xylem during water stress periods [43,64]. During the summer, the situations in Apulia and Liguria olive groves were clearly different: In Apulia, the number of spittlebugs collected in the olive agroecosystem during hot months was low, while in Liguria, spittlebugs were collected inside the olive groves in relatively high numbers until autumn. The continuative presence of spittlebugs on olive trees and in the agroecosystem is probably related to the less severe water stress of olive trees growing in Liguria compared to Apulia, Spain, and Greece, where adults almost disappear in summer from the olive agroecosystem [1,5,6,7,34]. Such a difference in crop exploitation may have important consequences if *X. fastidiosa* were introduced in Liguria, since the prolonged presence of vectors on the crop trees may contribute to the higher spread of the bacterium.

The relatively high number of spittlebugs on the herbaceous cover during autumn reflects the characteristic oviposition behavior of spittlebugs on weed stubble and dried vegetation [16]. It is a critical moment for disrupting the biological cycle of these insects, since the presence of old individuals, usually less motile than in the first months after emergence, may permit the application of control measures against adults more effective than in the rest of the year. Research on the drivers of oviposition preference by spittlebugs’ females could permit weed management of olive groves and surrounding habitats to obtain less attractive plant communities for oviposition in late summer, thus limiting the number of nymphs present inside or close to olive agroecosystems in the next generation.

Our data, together with those of [6,24,34,65], indicate that wild woody host-plants within and surrounding the olive agroecosystem may host a considerable number of spittlebugs, especially during summer, allowing aestivation of the spittlebugs. Some of the species naturally present in the olive agroecosystem and visited by spittlebugs are also hosts of *X. fastidiosa*, e.g., *Q. ilex*, *R. alaternus*, *Phillyrea latifolia,* and *Myrtus communis* [3]. These plant species may represent a reservoir of the bacterium, thus being potentially important for disease spread [56,57]. Accurate management of such wild woody plants, e.g., limiting their number inside or close to olive groves, may reduce the source of inoculum of *X. fastidiosa*, although the prevalence of transmission from olive to olive has been demonstrated [1].

## 5. Conclusions

This study provides a thorough description of host-plants selection and exploitation by spittlebugs in olive agroecosystems located in two Mediterranean regions of Italy, Apulia, and Liguria, representing different dry-summer climates and olive cropping conditions. The most important differences between the two areas concerned (i) host-plant association that, besides being determined by insect preference, was driven by ground cover floral composition, (ii) summer colonization of olive trees by adults, that visited olive canopies in a higher number and for a prolonged period in Northern Italy, and (iii) spittlebug population level that appeared to be higher in Northern Italy. Actually, several reports indicated that the populations of *P. spumarius* and of other spittlebug species are quite low under warm and dry Mediterranean conditions [6,7], although with some exceptions [8]. The knowledge of preferential plant selection by both nymphal and adult stages of the main *X. fastidiosa* vector in Europe may allow designing more effective integrated pest management programs, so far based on soil tilling to eliminate nymphs and on insecticide applications targeting adults on the crop canopy. The application of extensive soil tilling in dry environments such as some Mediterranean olive agroecosystems may compromise, in the long term, the quality of soil and augment the risk of desertification [66], while insecticides may impact on beneficial and non-target insects. Albeit an “aggressive” control strategy to limit the expansion of *X. fastidiosa* is nowadays needed, the future outlooks would be to implement low impact control methods, at least in the areas where eradication is no longer feasible. Examples of such control strategies are (i) targeted soil tilling/mowing of specific areas of the agroecosystems where spittlebugs nymphs tend to aggregate, e.g., plant communities rich in preferred host-plants, and (ii) shaping ground cover and wild plants composition within and close to susceptible crops to limit both colonization of spittlebugs and sources of inoculum of *X. fastidiosa*.

## Figures and Tables

**Figure 1 insects-11-00130-f001:**
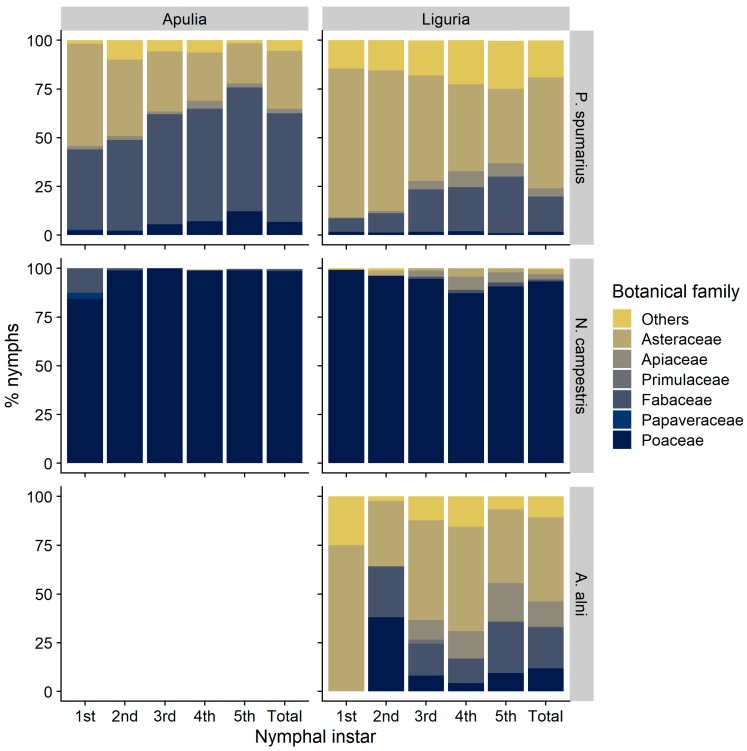
Relative percentage of nymphs of spittlebug species (*Philaenus spumarius*, *Neophilaenus campestris*, *Aphrophora alni*) on the most selected plant families in Apulia and Liguria olive groves.

**Figure 2 insects-11-00130-f002:**
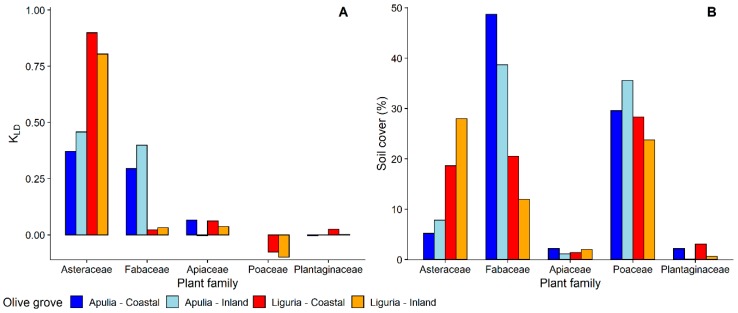
(**A**) Selection of most common plant families by nymphs of *Philaenus spumarius*—Kullback-Leibler divergence ($D_{KL}$) measure; (**B**) relative percentage of soil cover by most common plant families in Apulia and Liguria olive groves.

**Figure 3 insects-11-00130-f003:**
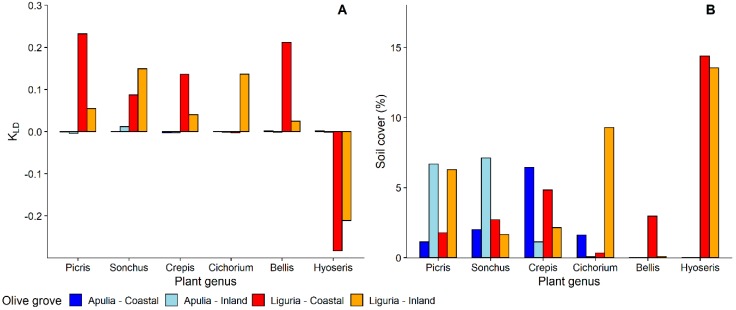
(**A**) Selection of most common plant genera of Asteraceae by nymphs of *Philaenus spumarius*—Kullback–Leibler divergence ($D_{KL}$) measure; (**B**) relative percentage of soil cover of most common plant genera of Asteraceae in Apulia and Liguria olive groves.

**Figure 4 insects-11-00130-f004:**
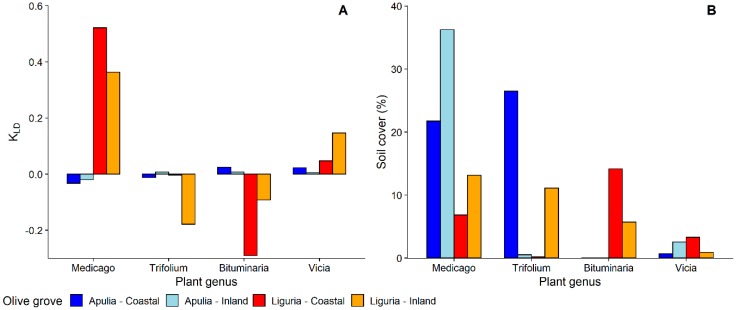
(**A**) Selection of most common plant genera of Fabaceae by nymphs of *Philaenus spumarius*—Kullback–Leibler divergence ($D_{KL}$) measure; (**B**) relative percentage of soil cover of most common plant genera of Fabaceae in Apulia and Liguria olive groves.

**Figure 5 insects-11-00130-f005:**
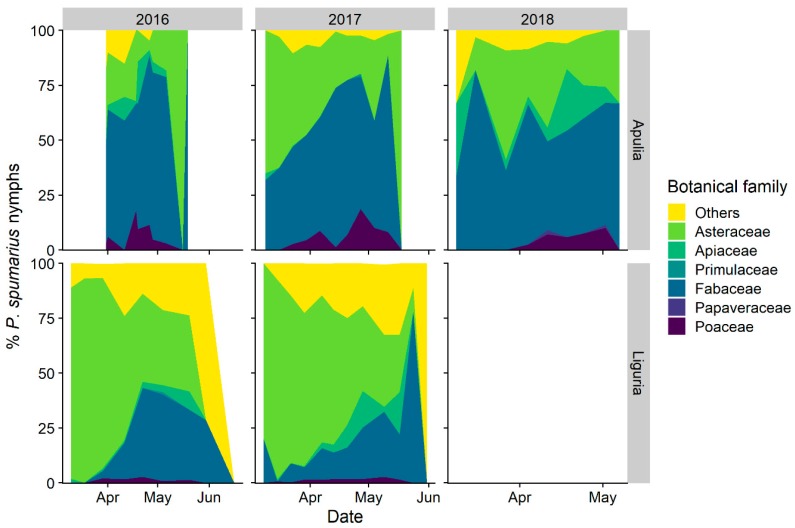
Relative percentage of nymphs of *Philaenus spumarius* on the most selected host-plant families (i.e., plant families hosting the highest number of spittlebug nymphs) in Apulia and Liguria olive groves during spring.

**Figure 6 insects-11-00130-f006:**
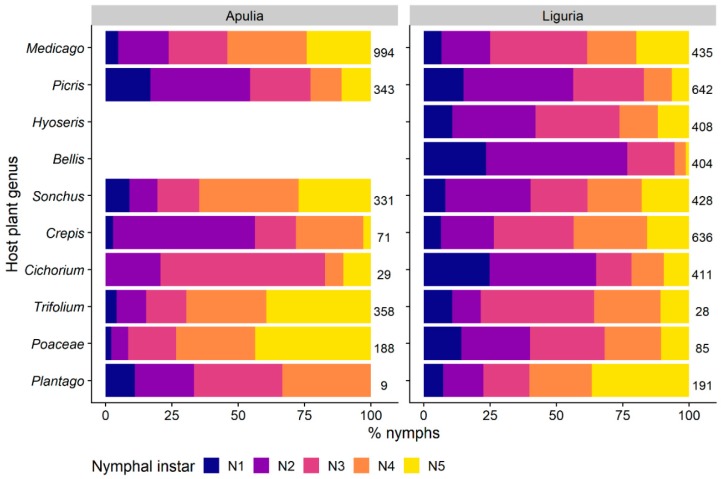
Relative percentage of the nymphal instars of *Philaenus spumarius* on the 10 most selected host-plant genera in Apulia and Liguria olive groves (i.e., plant genera hosting the highest number of spittlebug nymphs). Numbers beside bars represent the total number of nymphs sampled on each genus in each region.

**Figure 7 insects-11-00130-f007:**
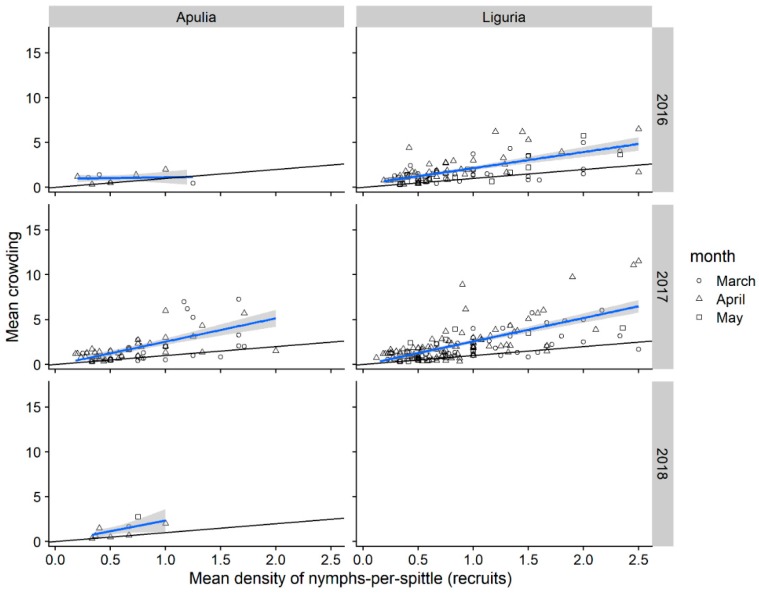
Relation of mean crowding to mean density of *Philaenus spumarius* nymphs-per-spittle in Apulia and Liguria olive groves. Blue line represents fitted values of Iwao’s regression. Black line represents x = y, expectation from Poisson series. Strong outlier data (>3rd quartile + 3× IQR) were discarded.

**Figure 8 insects-11-00130-f008:**
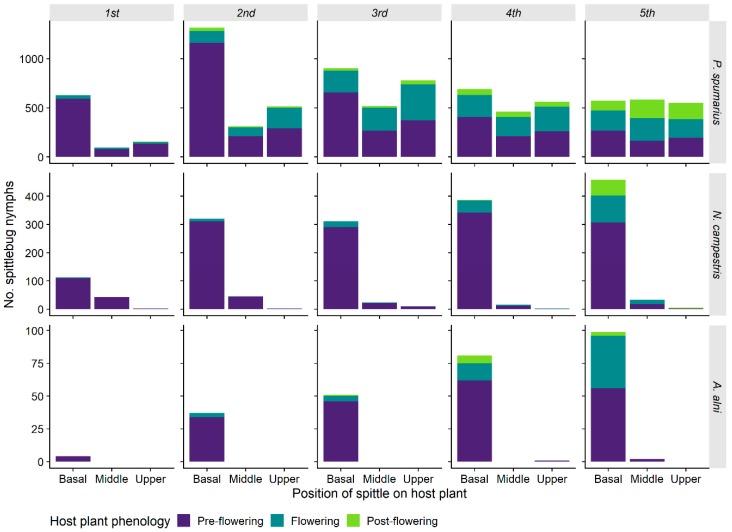
Abundance of spittlebugs nymphs (*Philaenus spumarius*, *Neophilaenus campestris*, *Aphrophora alni*) in relation to their instar, position on host-plant, and host-plant phenology.

**Figure 9 insects-11-00130-f009:**
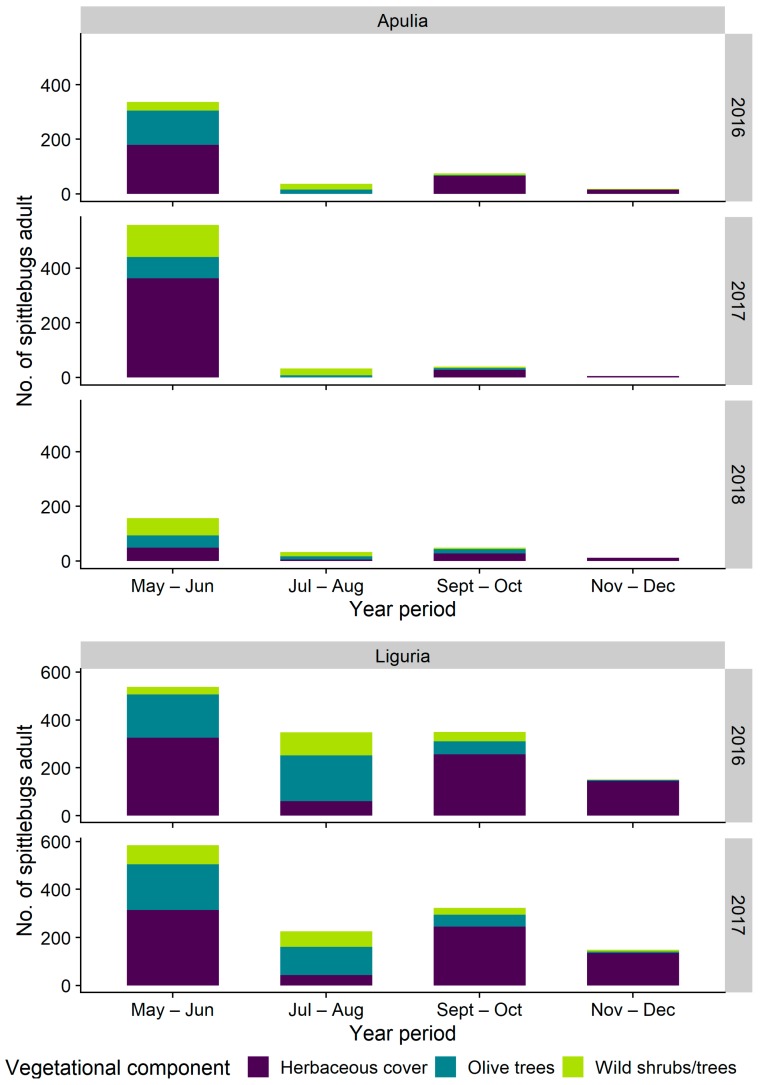
Abundance of *Philaenus spumarius* adults on different vegetational components in Apulia and Liguria olive groves throughout the year.

**Figure 10 insects-11-00130-f010:**
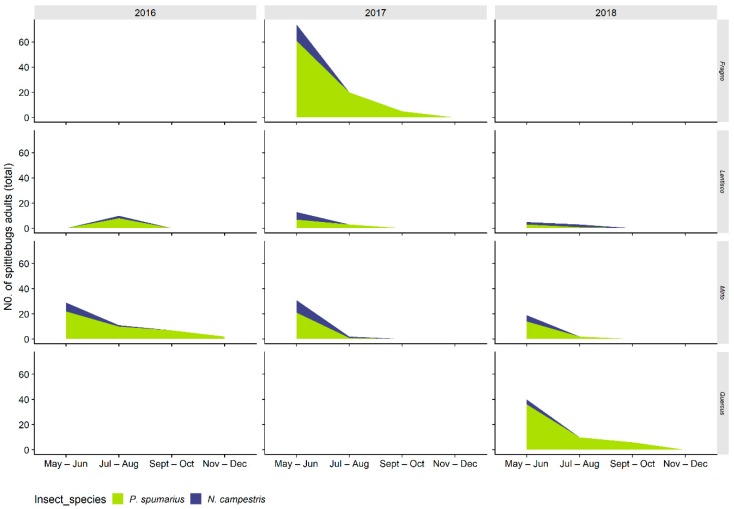
Abundance of spittlebug adults of different species (*Philaenus spumarius*, *Neophilaenus campestris*) on different wild woody plant species in Apulia olive groves throughout the year.

**Figure 11 insects-11-00130-f011:**
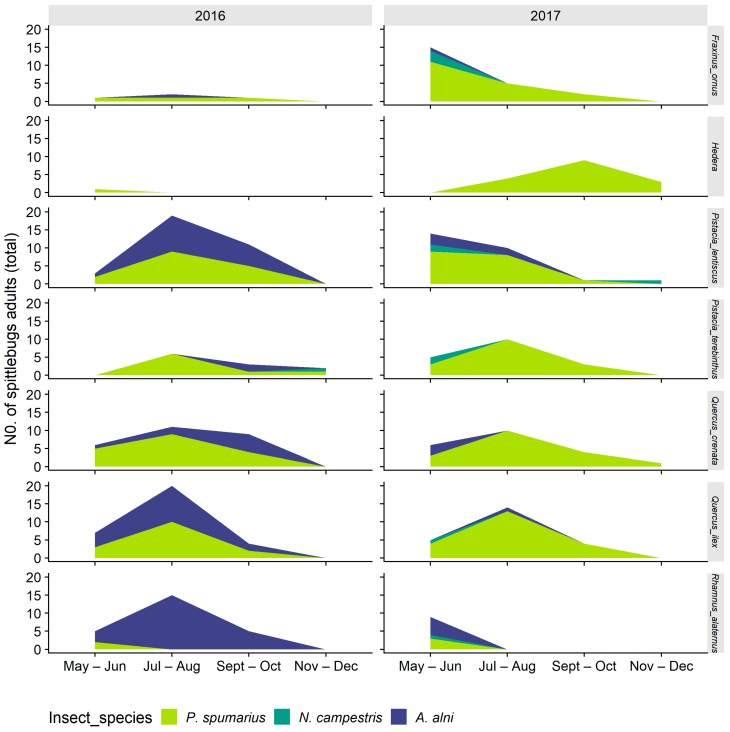
Abundance of spittlebug adults of different species (*Philaenus spumarius*, *Neophilaenus campestris*, *Aphorphora alni*) on different wild woody plant species in Liguria olive groves throughout the year.

**Figure 12 insects-11-00130-f012:**
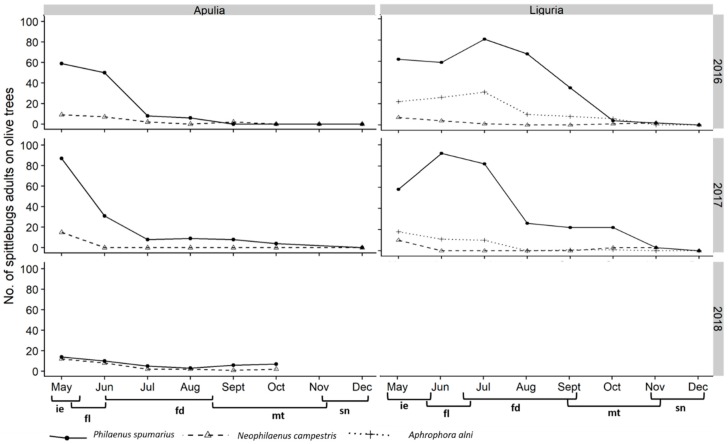
Abundance of spittlebug adults (*Philaenus spumarius*, *Neophilaenus campestris*, *Aphrophora alni*) on olive canopy in Apulia and Liguria olive groves throughout the year in relation to phenological stages of the crop, according to the BBCH scale (ie = inflorescence emergence; fl = flowering; fd = fruit development; mt = maturity of fruits; sn = senescence).

**Table 1 insects-11-00130-t001:** Iwao’s mean crowdedness regression parameters of nymphs of *Philaenus spumarius*.

	Apulia			Liguria	
	2016	2017	2018	2016	2017
**Nymps-per-spittle**					
Mean crowdedness (x)	0.107	1.18	0.229	0.812	0.758
Patchiness (P)	0.1	0.849	0.204	0.545	0.522
intercept	0.315 ± 0.475	−0.388 ± 0.175	−0.360 ± 0.424	−0.208 ± 0.119	−0.170 ± 0.116
slope	0.733 ± 0.259	2.526 ± 0.143	2.488 ± 0.190	2.244 ± 0.119	2.373 ± 0.118
R2 Adjusted	0.085	0.579	0.593	0.671	0.634
*p* value	0.144	>0.001	0.003	>0.001	>0.001
**Nymps-per-quadrat**					
Mean crowdedness (x)	1.93	5.02	3.67	8.08	10.1
Patchiness (P)	0.84	1.2	1.11	1.17	1.43
intercept	−3 ± 0.468	−1.41 ± 0.081	0.517 ± 0.331	0.064 ± 0.125	0.811 ± 0.103
slope	2.596 ± 1.304	1.722 ± 0.424	1.309 ± 1.289	1.541 ± 0.983	1.485 ± 0.951
R2 Adjusted	0.768	0.964	0.53	0.914	0.916
*p* value	0.001	>0.001	0.002	>0.001	>0.001

**Table 2 insects-11-00130-t002:** Effects of fixed covariates on the abundance of *Philaenus spumarius* nymphs. Poisson GLMM of nymphal counts with covariates spittle position, phenology host-plant, and instar, and their interactions. The estimated value for σ_Region_ was 0.64 and for σ_Year_ was 0.57.

Fixed Effects	χ^2^	*df*	*p*-Value
Spittle position	707.05	2	<0.001
Phenology host-plant	1403.3	2	<0.001
Instar	365.17	4	<0.001
Spittle position × henology host-plant	342.62	4	<0.001
Spittle position × instar	316.88	8	<0.001
Phenology host-plant × instar	971.1	8	<0.001
Spittle position × phenology host-plant × instar	121.5	16	<0.001
N	225		
AIC	4406.85		
R^2^ (fixed)	0.730

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
