# Peer review of "Spittlebugs of Mediterranean Olive Groves: Host-Plant Exploitation throughout the Year"

_insects, 2020, doi:10.3390/insects11020130_

Round 1
Reviewer 1 Report
Spittlebugs are common elements of the European fauna. However, they have gained major scientific interest only since they have been identified as vectors of the introduced bacterium Xylella fastidiosa in olive groves of southern Italy. Detailed knowledge of the ecological traits of those xylem sap-feeding insects like stage specific host plant affiliation and population structure is not only of scientific interest, but is also a prerequisite for the development of well-targeted control strategies against the bacterium.
The authors present a detailed report on the exploitation of host plants by immatures and adults of three spittlebug species, Philaenus spumarius, Neophilaenus campestris and Aphrophora alni. The first two species are confirmed vectors of X. fastidiosa in southern Italy. The study was carried out over two and three years in two Italian regions, Liguria and Apulia, respectively. The study focuses on the identification of larval host plants, feeding preferences, and spatial distribution patterns of the different instars between and on the host plants. In addition, the seasonal variation of the distribution of the three spittlebug species between ground cover, olive trees and wild woody plants was investigated. In studies like this one, which include experimental plots in different regions with different ecoclimatic conditions, the comparison of standardized data is a particular challenge. The authors have solved this problem with regard to larval host preference by applying the Kullback-Leibler statistics, which allows them to estimate host preference independently of the variation in frequency of individual host plant species. In general, the study presents new data, which are important not only for a better understanding of the spittlebug ecology but also for the setup of appropriate control strategies against the vectors of X. fastidiosa. The experimental approach is well planned, the statistical treatment is sound and the conclusions are well based on the experimental results. There are some minor points that should be corrected before publication. I therefore recommend acceptance of the paper after minor revision.
- There is text missing between line 210 and 211: At least the headings of sections 3 and 3.1 need to be inserted.
- Line 260: Replace “Figure 2” by “Table S1”. The figure 2 does not refer to the plant genera mentioned in this sentence.
- Table 1: The year dates are not located above the corresponding columns.
- Table S1: Add “spp.” to “Fabaceae” in cell B36
Author Response
Reviewer #1
Comments and Suggestions for Authors
Spittlebugs are common elements of the European fauna. However, they have gained major scientific interest only since they have been identified as vectors of the introduced bacterium Xylella fastidiosa in olive groves of southern Italy. Detailed knowledge of the ecological traits of those xylem sap-feeding insects like stage specific host plant affiliation and population structure is not only of scientific interest, but is also a prerequisite for the development of well-targeted control strategies against the bacterium.
The authors present a detailed report on the exploitation of host plants by immatures and adults of three spittlebug species, Philaenus spumarius, Neophilaenus campestris and Aphrophora alni. The first two species are confirmed vectors of X. fastidiosa in southern Italy. The study was carried out over two and three years in two Italian regions, Liguria and Apulia, respectively. The study focuses on the identification of larval host plants, feeding preferences, and spatial distribution patterns of the different instars between and on the host plants. In addition, the seasonal variation of the distribution of the three spittlebug species between ground cover, olive trees and wild woody plants was investigated. In studies like this one, which include experimental plots in different regions with different ecoclimatic conditions, the comparison of standardized data is a particular challenge. The authors have solved this problem with regard to larval host preference by applying the Kullback-Leibler statistics, which allows them to estimate host preference independently of the variation in frequency of individual host plant species. In general, the study presents new data, which are important not only for a better understanding of the spittlebug ecology but also for the setup of appropriate control strategies against the vectors of X. fastidiosa. The experimental approach is well planned, the statistical treatment is sound and the conclusions are well based on the experimental results. There are some minor points that should be corrected before publication. I therefore recommend acceptance of the paper after minor revision.
- There is text missing between line 210 and 211: At least the headings of sections 3 and 3.1 need to be inserted.
Done, thank you for pointing out this error.
- Line 260: Replace “Figure 2” by “Table S1”. The figure 2 does not refer to the plant genera mentioned in this sentence. Right, done
- Table 1: The year dates are not located above the corresponding columns. Corrected
- Table S1: Add “spp.” to “Fabaceae” in cell B36. Done
Reviewer 2 Report
In order to support the conclusion, I suggest that the authors provide one figure in the content or appendix for the report, which shows the olive phenology, the annual appearance of both of adults and nymphs for target spittlebug species, and the farmer's year-round cultivation strategy. This will help readers better understand the highlights of this report.All of confirmed insect vectors of Pierce’s disease of grape in Taiwan are sharpshooters. Although different spittlebug species occur in the epidemic areas, they cannot be proved by Koch’s postulates. In
Taiwan, farmers use lawn mowers to control weed hosts of vector insects, which has greatly reduced grape disease.
Author Response
Reviewer #2
Comments and Suggestions for Authors
In order to support the conclusion, I suggest that the authors provide one figure in the content or appendix for the report, which shows the olive phenology, the annual appearance of both of adults and nymphs for target spittlebug species, and the farmer's year-round cultivation strategy. This will help readers better understand the highlights of this report.
Some of the authors already produced a schematic diagram which shows life cycle of Philaenus in relation with host plants in olive agroecosystems. This is published in EFSA (European Food Safety Authority), Vos S, Camilleri M, Diakaki M, Lázaro E, Parnell S, Schenk M, Schrader G, and Vicent A, 2019. Pest survey card on Xylella fastidiosa. EFSA supporting publication 2019:EN-1667. 53 pp. doi:10.2903/sp.efsa.2019.EN-1667. This source has been cited in the introduction for readers’ convenience.
All of confirmed insect vectors of Pierce’s disease of grape in Taiwan are sharpshooters. Although different spittlebug species occur in the epidemic areas, they cannot be proved by Koch’s postulates. In
Taiwan, farmers use lawn mowers to control weed hosts of vector insects, which has greatly reduced grape disease
In Italy, soil tilling has been noticed as more effective than mowing in suppressing nymph population, and therefore this measure has been recommended (Italian Ministerial Decree 2180). However, the discussion of the efficacy of different control measures, is beyond the scope of our manuscript, which deals with host-plant exploitation by spittlebugs.
Reviewer 3 Report
see file

Author Response
Reviewers #3
The authors have done a great job in determining the host range of spittlebugs, the relative
attractiveness of certain plants to spittlebugs and the aggregation of spittlebugs over plants.
There is however a discrepancy between what the authors explain as statistical analysis in the
Methods section and what is reported in the results section. Moreover, some sentences are confusing/not
understandable.
comments
L 33, 97: The authors say that their research will allow to assess epidemiological risks, but I do not
See how. At 3 places in the discussion (L456-457, L486-487, L514-517) the authors do some suggestions.
In the light of the current spread of Xylella this kind of reasoning seems to be crucial to put into a
conclusion or abstract.
Thank you for the suggestion. Abstract and Conclusions have been implemented accordingly.
Methods section about the statistical analysis: L162-L173
The methods section mentions a negative binomial GLMM (where do you give the results of this
analysis?). In the results section only a Poisson GLMM is mentioned in the header of table 1, and there
the covariates and the response variable are, in my view, different from the ones mentioned in this
part.
Thank you for pointing out these mistakes. There was a formatting error in the MS sent to the reviewers, and the first part of the results was joined to the last part of M&M, hence hiding the explanation of negative binomial GLMM results. In the first part of Results now is mentioned and showed negative binomial GLMM analysis. Also, the explanation in M&M of Poisson GLMM was missing, also due to an oversight. It has been added.
It is not clear what the authors consider the response variable; is that the number of spittlebug
nymphs? It is not clear what the authors consider the covariates; are that cover percentage and mean height?
Correct, now it has been written more clearly.
In L167 the authors state “percentage distribution”, but I do not know what that is. Do the authors
compare the distribution of fraction of plant species where P.s. is found with the distribution of the
fractions of the areas where the plant species where found?
The two step analysis as explained in L179-184 is not clear to me either: in step 1 you should start
With the plant species you found at the sampling times and tell what you did with that, and I presume that
you averaged it thereafter (but it remains unclear from the text what you did). The same holds for the
other so-called “mean distribution” in the first step.
It was indeed not very clear that part. The explanation of the KLD analysis has been substantially rewritten to clarify the concepts. Basically, we compared the percentage of cover of one plant taxon with the percentage of nymphs of P.s. found on that plant taxon. See the new version of MS for the detailed explanation.
The mean crowding in L198 is a non-linear function of the mean of . Why can that be estimated
With a linear equation as in L203?
To our knowledge mean crowding is linearly related to mean density over a wide range of situations in insects (different densities, stages, species); “If mean crowding is plotted against mean density, the relation can be fitted to a linear regression in a wide variety of situations, including both theoretical distribution models and population samples of insects” (Iwao 1968; Southwood and Henderson 2000). In theory the relationship could be curvilinear with s2 (variance) lower than mean, different from what recorded in our case. However, even in underdispersed populations (s2 < mean) no curvilinear relationships were found by Iwao (1968, 1977). Please see References where we reported extracts of cited papers relevant to clarify these concepts.
L208 You do not mean to write uniform distribution because the statistical definition of uniform
distribution on an interval (a, b) is that every number in the interval is as likely as another one. In
your case you might want to refer to a distribution that is more regular than Poisson. Absolutely correct, we changed that sentence. Thank you for pointing it out.
References
Southwood R, Henderson P (2000) Ecological Methods
“Clarification and unification of these various approaches were achieved by Iwao (1968, 1970a,b,c, 1972), who initially demonstrated that the relationship of meancrowding, x., to mean density, x, for a species could be expressed over a range of den sities by a linear regression:
= a + bx”
Iwao S (1968) A new regression method for analyzing the aggregation pattern of animal populations. Researches on Population Ecology 10:1–20. https://doi.org/10.1007/BF02514729
“… we can see that in every example a linear regression can be, at least approximately, fitted to the
observed relation. In some other cases examined, the points are scattered more irregularly, perhaps due to sampling errors or heterogeneous nature of the data, but none of the cases is found to be fitted to a curvilinear regression.”
“From both theoretical consideration and the examination of actual examples, it can be said that the regression of mean crowding on mean density is linear in a wide variety of situations. The intercept r and slope/9 of the regression equation (4) seem to be useful as indices describing different aspects of dispersion pattern of populations of a given species (and stage).”
“If mean crowding is plotted against mean density, the relation can be fitted to a linear regression in a wide variety of situations, including both theoretical distribution models and population samples of insects.”
Iwao S (1977) Analysis of spatial association between two species based on the interspecies mean crowding. Population Ecology 18: 243-260.
Round 2
Reviewer 3 Report
all things that were unclear are changed in this revision